# The Binding of Monoclonal and Polyclonal Anti-Z-DNA Antibodies to DNA of Various Species Origin

**DOI:** 10.3390/ijms22168931

**Published:** 2021-08-19

**Authors:** Diane M. Spencer, Angel Garza Reyna, David S. Pisetsky

**Affiliations:** 1Department of Medicine and Immunology, Division of Rheumatology, Duke University Medical Center, Durham, NC 27710, USA; diane.m.spencer@duke.edu; 2Medical Research Service, Veterans Administration Medical Center, Durham, NC 27705, USA; 3Department of Chemistry, Duke University, Durham, NC 27705, USA; angel.garza.reyna@duke.edu

**Keywords:** DNA, Z-DNA, B-DNA, immunoassay, anti-DNA antibodies, conformation, helix, phosphodiester backbone

## Abstract

DNA is a polymeric macromolecule that can display a variety of backbone conformations. While the classical B-DNA is a right-handed double helix, Z-DNA is a left-handed helix with a zig-zag orientation. The Z conformation depends upon the base sequence, base modification and supercoiling and is considered to be transient. To determine whether the presence of Z-DNA can be detected immunochemically, the binding of monoclonal and polyclonal anti-Z-DNA antibodies to a panel of natural DNA antigens was assessed by an ELISA using brominated poly(dG-dC) as a control for Z-DNA. As these studies showed, among natural DNA tested (*Micrococcus luteus,* calf thymus, *Escherichia*
*coli*, salmon sperm, lambda phage), micrococcal (MC) DNA showed the highest binding with both anti-Z-DNA preparations, and *E. coli* DNA showed binding with the monoclonal anti-DNA preparation. The specificity for Z-DNA conformation in MC DNA was demonstrated by an inhibition binding assay. An algorithm to identify propensity to form Z-DNA indicated that DNA from *Mycobacterium tuberculosis* could form Z-DNA, a prediction confirmed by immunoassay. Together, these findings indicate that anti-Z-DNA antibodies can serve as probes for the presence of Z-DNA in DNA of various species origin and that the content of Z-DNA varies significantly among DNA sources.

## 1. Introduction

DNA is a polymeric macromolecule whose structural diversity is essential for its role in heredity and gene expression [1]. In solutions of DNA at physiological salt concentrations, the predominant structure is B-DNA. B-DNA is a right-handed double helix with classical Watson–Crick base pairing. A variety of other helical conformations have been identified initially with synthetic molecules of a defined base sequence [2]. Of these conformations, Z-DNA exists as a left-handed helix in which the phosphodiester backbone has a zig-zag structure with bases in alternating *syn* and anti conformations [3,4,5]. The unique structural features of Z-DNA have suggested a role in the regulation of transcription, although this role has not been well defined [5,6,7,8].

As with other non-B-DNA conformations, Z-DNA depends on the base sequence, base modification and ionic conditions. Supercoiling can also influence the display of this conformation [9,10,11]. As shown using X-ray diffraction and other physical–chemical techniques, alternating guanosine cytosine (GC) sequences can transition to Z-DNA [3]. Whereas poly(deoxyguanylic–deoxycytidylic) acid [poly(dG-dC).poly(dG-dC)], further abbreviated in this paper to [poly(dG-dC)], can undergo a B- to Z-DNA transition at high-salt conditions, bromination of poly(dG-dC) [Br-poly(dG-dC)] produces a synthetic DNA in which the Z-DNA conformation is present at low-salt conditions [3]. Algorithms to predict the presence of Z-DNA on the basis of thermodynamic considerations indicate that a variety of sequences, and not just alternating pyrimidine–purine tracts, can form Z-DNA in the genome [12,13,14,15,16].

In addition to physical–chemical techniques, Z-DNA can be recognized immunochemically by specific antibodies. Unlike B-DNA, which is generally nonimmunogenic, Z-DNA can elicit a robust antibody response by experimental immunization of animals with Br-poly(dG-dC). Base modification by acetylaminofluorene also produces a stable form of Z-DNA that is immunogenic [17,18,19,20,21,22,23]. Antibodies to Z-DNA also occur spontaneously in patients with systemic lupus erythematosus (SLE) and animal models of this disease [24,25,26,27,28]. SLE is a prototypic autoimmune disease characterized by antibodies to DNA and other nuclear macromolecules. Anti-Z antibodies obtained from immunized mice or autoimmune MRL-*lpr* mice have provided valuable reagents to identify Z-DNA in a number of settings, including polytene chromosomes of Drosophila and cultured cells [29,30,31,32,33]. While these studies provide evidence that Z-DNA can occur in chromosomal DNA, proteins are present in these settings, and fixatives may influence the conformation of DNA in situ.

In the current study, we explored another approach to identify Z-DNA using ELISA assays with polyclonal and monoclonal anti-Z-DNA antibodies, investigating whether Z-DNA is present antigenically in naturally occurring DNA of various species origin. This approach differs from prior studies, as the DNA is purified and, therefore, lacks proteins that could influence the presence of Z-DNA. Furthermore, we assayed for the binding of anti-Z-DNA antibodies under ordinary salt conditions (i.e., 150 mM NaCl), as well as in the absence of chemical agents that can affect the B- to Z-DNA transition. As the data reported herein show, anti-Z-DNA antibodies can bind differentially to naturally occurring DNA from various species, although the extent of the binding varies markedly. These findings provide the basis for an immunochemical approach to identify Z-DNA with purified DNA that can be used in a complementary way with predictive algorithms for structural analysis.

## 2. Results

In these studies, we used two sources of antibodies that are commercially available: a polyclonal sheep antiserum elicited by immunization with Br-poly(dG-dC) and a monoclonal anti-DNA antibody obtained by fusion of immune cells of an autoimmune MRL-*lpr* mouse. As a source of Z-DNA, we used poly(dG-dC), which was brominated by treatment with bromine water, to provide a stable source of Z-DNA [Br-poly(dG-dC)]. In contrast to previous studies [17,34], we found that bromination at 150 mM NaCl was more effective at producing Z-DNA than higher salt concentrations, as determined by optical density at 260 nm and 295 nm. Because the product from bromination at 150 mM showed strong reactivity with both the monoclonal and polyclonal anti-Z-DNA antibody preparations, we believe that the preparation of Br-poly(dG-dC) we used is, in fact, Z-DNA.

For DNA antigen, initial studies involved a series of DNA preparations obtained commercially. These DNA included human placenta (HP), *Escherichia coli* (EC), salmon sperm (SS), calf thymus (CT), *Micrococcus luteus* (MC) and lambda. The base composition of these DNA antigens is provided in Table 1. Among these DNA, MC DNA is notable because of its high GC content; furthermore, specific antibodies to MC DNA occur in the blood of otherwise healthy individuals. These antibodies differ from anti-DNA antibodies in patients with SLE, which are broadly reactive with DNA because of binding to the phosphodiester backbone [35,36]. These findings suggest that MC DNA bears sequences or conformations that can induce specific antibodies as part of infection or colonization.

Finally, we performed an experiment to help determine whether the Z-DNA structure is pre-existent in MC DNA or whether it results from a transition that takes place during the course of the incubation with the antibody preparation. We hypothesized that, with a transition from B to Z-DNA, antibody binding to MC DNA would take place more slowly than with Br-poly(dG-dC) and depend on the rate coefficient of any transition.

In these experiments, we assessed antibody binding to DNA in various formats. The first format involved the assessment of direct antibody binding in an ELISA in which DNA preparations were coated to the plates at varying concentrations and then reacted with the anti-Z-DNA antibody preparations at a single dilution. This dilution was established by prior titration experiments using poly(dG-dC) and Br-poly(dG-dC) as test antigens. As a control, the binding of an SLE plasma containing anti-DNA was also tested. This plasma was selected because it has a very high titer of antibodies to CT DNA and other natural DNAs studied. No DNA and unbrominated poly(dG-dC) antigen conditions were included as negative controls. These conditions yielded very low reactivity (OD 450 nm 0.05–0.5) with test antibodies, plasma and antiserum.

Figure 1 presents the results of these studies. As these data indicate, both the monoclonal and polyclonal anti-Z-DNA antibody preparations bound well to Br-poly(dG-dC) but did not bind to poly(dG-dC), which was not brominated. With natural DNA, both antibody preparations showed significant binding to MC DNA. While the monoclonal antibody bound well to EC DNA, the polyclonal preparation, however, showed much less reactivity. The SLE plasma bound similarly to all of the natural DNA in the panel indicating that, under the conditions used for the ELISA, similar amounts of DNA were available for antibody binding. As such, differences in the reactivity of the anti-DNA preparation cannot be attributed to the amount of DNA on the plate.

As another approach to determine the presence of Z-DNA structure, and to confirm the results in Figure 1, we then tested the reactivity of the antibody preparations at varying dilutions with a fixed concentration of DNA on the plate. Because a polyclonal preparation may have antibodies to a variety of Z-DNA determinants, the titration of the antibody provides another perspective on DNA antigenicity.

As shown in Figure 2, the greatest reactivity with the monoclonal antibody was observed with MC DNA, EC DNA and the Br-poly(dG-dC). Both lambda DNA and CT DNA showed some reactivity with the monoclonal antibody at low dilutions, whereas the Br-poly(dG-dC) and MC DNA had plateau binding even at a titer of 1:102,400. The titer for EC DNA was lower, although, even at a dilution of 1:51,200, binding was still about half maximal.

The results with the polyclonal antibody were similar, with both Br-poly(dG-dC) and MC DNA having significant binding at dilutions of 1:3200. At lower antibody dilutions, EC DNA and SS DNA showed binding comparable to that observed with MC DNA. Together, these experiments confirm the much greater reactivity of MC DNA with anti-Z-DNA preparations.

Unlike the monoclonal anti-Z-DNA antibody, which has a single specificity, the polyclonal anti-Z-DNA preparation potentially has antibodies to more than one Z-DNA determinant or even B-DNA. In view of the high titer of this antiserum, we thought that it was important to show that the determinant recognized was, in fact, Z-DNA. Therefore, we performed inhibition experiments to test the ability of Br-poly(dG-dC) to inhibit binding to MC DNA.

As shown in Figure 3, Br-poly(dG-dC) effectively inhibited binding to MC DNA at various antiserum dilutions; in contrast, unbrominated poly(dG-dC) was ineffective as an inhibitor.

DNA antigens used in this study were a convenience sample that could be purchased commercially in quantity. In view of the findings with MC DNA, we wanted to test DNA from other bacterial species, especially those that would be clinically relevant. We, therefore, chose DNA from *Mycobacterium tuberculosis* (MTb), as it has a high GC content. A limited analysis using the *Z-Hunt-II* program indicated that MTb DNA would also be predicted to have an increased content of sequence with the potential to form Z-DNA (Table 1).

As shown in Figure 4, MTb DNA shows significant reactivity with both the monoclonal and polyclonal anti-Z-DNA antibody preparations. With the polyclonal antibody preparation, binding was less than that of MC DNA although much greater than CT DNA. These findings suggest that DNA from a number of species can display Z-DNA structures.

Figure 5 provides these results. As these data indicate, the time course of antibody binding to MC DNA and Br-poly(dG-dC) was similar. Binding to either EC DNA or CT DNA was low but also did not change over time. Together, these data suggest the presence of the Z-DNA structure in MC DNA does not result from a time-dependent transition during incubation.

## 3. Discussion

These studies provide new insight into the use of antibodies to detect the presence of the Z-DNA conformation and show that both a monoclonal and polyclonal antibody preparation can bind to purified DNA of various species origin in an ELISA. Furthermore, these studies indicate that the extent of antibody binding varies with the source of DNA, with MC DNA showing the highest level of antibody reactivity among the DNA initially tested. This reactivity is not confined to MC DNA, however, as MTb DNA was also antigenically active. The simplicity of an ELISA, coupled with an ability to perform both direct binding and inhibition binding assays, provides a valuable new approach to probe DNA structure by immunochemical assays.

In prior studies, antibodies to Z-DNA have been used to show the presence of Z-DNA in chromosome preparations or cells, providing strong evidence that Z-DNA structure occurs naturally and is not just a feature of synthetic DNA molecules or nonphysiological conditions (i.e., high salt) [29,30,31,32,33]. In systems for investigating the presence of Z-DNA in situ, proteins are present and could promote or stabilize a transition to Z-DNA. An effect of fixatives is also possible. In contrast to this prior work, our studies involved purified DNA and assays conducted under physiological salt conditions. The binding of antibodies to Br-poly(dG-dC) in inhibition assays indicates, moreover, that the presence of Z-DNA does not depend on any interaction with a solid-phase support.

Previous studies have characterized the binding of antibodies to Z-DNA, with chemically modified synthetic DNA the usual target antigen [17,18,19,20,21,22,23,24,25,26,27,28]. These studies have included natural DNA as a control, with most studies indicating the lack of reactivity of natural DNA with anti-Z-DNA antibodies presumably because the DNA existed primarily in the B-DNA conformation. Our findings are not inconsistent with those studies, as we found that some of the natural DNA antigens were indeed much less reactive than Br-poly(dG-dC). Our findings differ from the prior work, however, as we studied a broader range of DNA sources, including MC DNA, which has both a high GC content and predicted Z-DNA-potentiality. Preliminary analysis with the *Z-Hunt-II* program also suggests that MC DNA has a higher predicted content of Z-DNA than all other DNA in the original group tested. The situation is the same for MTb DNA, which also has a high GC content.

Our studies differ from prior studies in other respects. Thus, we coated plates directly with DNA preparations and did not use a coating agent such as poly-L-lysine, which can increase solid-phase binding but could also potentially affect the conformation of DNA. We also brominated the poly(dG-dC) at 150 mM NaCl rather than 4 M, as performed in previous studies [17,34]. We do not know why we could generate Z-DNA at low-salt conditions, although it is possible that our poly(dG-dC) preparation differed in some way from those in the previous experiments. The original studies using Br-poly(dG-dC) as a model for Z-DNA were performed for many years, and we can speculate that changes in the methodology for the synthesis of polymers may affect the actual amount of alternating dGdC sequence in a random polymer. The size of the polymer may also affect the tendency to form Z-DNA. These issues are under investigation.

While our preparation of Br-poly(dG-dC) may have differed from that of prior studies in the conditions for incubation with bromine water, it bound well to both the monoclonal and polyclonal anti-Z-DNA antibodies. Importantly, these antibodies provided consistent results with Br-poly(dG-dC) and the reactive DNA antigens such as MC DNA. In this regard, we observed that Br-poly(dG-dC) preparations showed diminished reactivity in the ELISA over time, suggesting a loss of the Z-DNA conformation. We, therefore, prepared the brominated Z-DNA several times to conduct these experiments. This variability limited experiments to calculate more precisely the amount of Z-DNA structure in natural DNA. In contrast, the pattern of reactivity of the natural DNA was much more consistent over time. We do not have information concerning the stability of Br-poly (dG-dC) prepared under high-salt conditions.

In these studies, the polyclonal and monoclonal anti-Z-DNA antibody preparations produced generally similar results. Thus, both antibody preparations bound well to Br-poly(dG-dC) and did not react with unbrominated poly(dG-dC). These antibody preparations also both bound well to MC DNA and MTb DNA, although some differences in the extent of binding were noted with some of the other natural DNA. In particular, the monoclonal anti-Z-DNA showed more reactivity to EC DNA than the polyclonal preparation. Previous studies have indicated that monoclonal anti-DNA antibodies can differ in their pattern of reactivity with Br-poly(dG-dC), suggesting differences in the structural features of Z-DNA recognized [17,18,20,21,22,27]. At present, only a single monoclonal anti-DNA is commercially available, limiting more precise determination of fine specificity. It should be noted that while a polyclonal anti-Z-DNA antibody preparation may contain multiple specificities, only one or a few of the component specificities may contribute to the binding observed when used at high dilution. As such, the extent to which polyclonality affects binding very much depends on the assay format and the relative titer of the different component specificities.

While an algorithm can predict the presence of Z-DNA in genomic DNA, the demonstration of antibody binding in natural DNA is perhaps surprising, especially under physiological salt conditions. In general, Z-DNA has been conceptualized as a transient conformation whose presence can be promoted by, among other factors, salt conditions, base modification, supercoiling and protein binding [3,4,5]. The simplest explanation for our findings is that regions of the DNA with the potential to form Z-DNA readily undergo a B- to Z-DNA transition under the assay conditions we used. Once in a Z-DNA conformation, antibody binding occurs rapidly, stabilizing the conformation. In this regard, our studies indicate that the binding of the anti-Z-DNA antibodies to unbrominated poly(dG-dC) was, at most, very limited, even though an alternating dG-dC sequence has a high potential for a B- to Z-DNA transition. Therefore, it is possible that the rate constant for a B- to Z-DNA transition is higher in natural DNA than synthetic DNA.

In addition to the base sequence, the display of Z-DNA also depends on base methylation, a key structure for epigenetic regulation. The pattern of methylation differs among species; however, whereas cytosine is commonly methylated to 5-methylated cytosine (5mC) in CpG dinucleotides in mammalian organisms, methylation of adenine is the predominant modification in bacteria [37,38,39]. These differences may impact the content of Z-DNA. As shown using synthetic polymers, the presence of 5mC can promote a transition to Z-DNA depending on the concentration of salt and divalent cations. Like high concentrations of sodium chloride, MgCl_2_ at high concentrations (i.e., 0.7 M) can lead to the formation of Z-DNA, although with polymers with 5mC, the transition occurs at concentrations of MgCl_2_ orders of magnitude lower than those needed for polymers with unmodified cytosine [40,41,42,43]. It is therefore possible that, even at low concentrations of MgCl_2_, the binding of anti-Z-DNA antibodies could be affected in the ELISA. In this scenario, an increase in antigenicity would be expected with mammalian DNA because of the much greater content of 5mC than present in bacterial DNA.

Many studies have now demonstrated the presence of Z-DNA sequences in natural DNA by a variety of analytic approaches, with the presence of Z-DNA-forming sequences in certain locations in the genome suggesting a role of Z-DNA in regulating gene expression [6,7,8]. There has also been an interest in how Z-DNA can transiently arise as the RNA polymerase molecule progresses along the DNA during transcription. A role of Z-DNA in host defense has also been postulated in view of the expression of a protein called ZBP1 which binds Z-DNA as well as Z-RNA [44,45,46]. Therefore, it is of interest that, among the DNA with the highest content of Z-DNA immunochemically defined, two of the DNA were of bacterial origin, raising the possibility that the presence of Z-DNA in bacterial DNA can influence host defense.

Having demonstrated the feasibility of using antibodies to detect Z-DNA by ELISA, we are now conducting experiments to assess the content of Z-DNA in other bacterial, viral and fungal DNA. We hope that experiments of this kind will provide further insight into the role of Z-DNA in the regulation of gene expression, as well as immune cell stimulation in infectious and autoimmune diseases.

## 4. Materials and Methods

### 4.1. DNA Antigens

DNA from *Micrococcus luteus* (MC), *Escherichia coli* (EC), human placenta (HP), salmon sperm (SS) and lambda were purchased from Sigma-Aldrich (St. Louis, MO, USA). Calf thymus (CT) DNA was purchased from Worthington Biochemical Corp. (Lakewood, NJ, USA). *Mycobacterium tuberculosis* (MTb) DNA was the kind gift of Dr. David Tobin from Duke University. CT and MC DNA preparations were further purified by phenol/chloroform extraction and showed OD 260/280 ratios of 1.7–1.8. Other DNA antigens were used as provided by the manufacturer.

### 4.2. Production of Z-DNA

To produce Z-DNA, we brominated poly(deoxyguanylic–deoxycytidylic) [poly(dG-dC)], which was purchased from Sigma-Aldrich. Poly(dG-dC) is a double-stranded, alternating copolymer that exists as B-DNA when unbrominated and as Z-DNA when brominated. The bromination protocol used was adapted from published methods (17,34) and was as follows: poly(dG-dC) was reconstituted to 500μg/mL with 1X TE buffer (10 mM Tris-HCl, pH 8.0, 1 mM EDTA. Both reagents from Sigma-Aldrich). The sodium content of an aliquot of 500 μg/mL was adjusted to 150 mM. This stock was then diluted to 200 μg/mL with citrate/EDTA/NaCl buffer (20 mM sodium citrate buffer, pH 7.2; 1 mM EDTA and 150 mM NaCl). Bromine water (30 g/L; Hach Company, Loveland, CO, USA) was diluted to 1/25 with UltraPure Distilled Water (Invitrogen-ThermoFisher, Waltham, MA, USA). The diluted bromine water and poly(dG-dC) were incubated at room temperature (RT; 19–23 °C) for 20 min at a ratio of 1.3:1 by volume.

After the incubation, the reaction mix was dialyzed against 1X TBS (15 mM Tris-HCl, pH 7.4; 150 mM NaCl and 1 mM EDTA) using G-Biosciences Tube-o-Dialyzer devices (St. Louis, MO, USA). The absorbance at 260 nm and 295 nm was determined using a NanoDrop™ 1000 spectrophotometer (Invitrogen-ThermoFisher, Waltham, MA, USA). Then, 1X TBS was used as a blank. A ratio of 260 nm absorbance/295 nm absorbance can be used to determine the extent of the bromination of poly(dG-dC) and the formation of Z-DNA. Poly(dG-dC) in the B-DNA form gives ratio values of between 7 and 11. The ratio for the brominated form is around 2–4. In this laboratory, Br-poly(dG-dC) form was found to be stable for up to seven days at 4 °C.

### 4.3. ELISA Assays

ELISA assays were performed as described previously [47]. Briefly, Immulon 2HB (Invitrogen-ThermoFisher) plates were coated overnight with 100 μL/well of DNA from various sources diluted in 1X SSC (150 mM NaCl, 15 mM sodium citrate, pH 7.0). Control wells without DNA (1X SSC alone) were included. Coated plates were incubated overnight at 4 °C. On the next day, plates were washed 3X with 1X Phosphate Buffered Saline (PBS) followed by blocking for 2 h at RT with block buffer (2% bovine serum albumin (BSA), 0.05% Tween-20 in 1X PBS). After blocking, plates were washed 3X with PBS. The ELISA plates were then incubated for 1 h at RT with diluted sources of anti-Z-DNA antibodies: mouse monoclonal anti-Z-DNA antibody (Absolute Antibody, Boston, MA, USA), sheep polyclonal anti-Z-DNA antibody (Abcam; Cambridge, MA, USA), or human SLE plasmas (Plasma Services Group; Huntingdon Valley, PA, USA). All antibodies were diluted with Tris ELISA dilution buffer (0.1% BSA, 0.05% Tween 20 in 50 mM Tris, pH 7.4).

After incubation with antibodies, the ELISA plates were washed 3X with 1X PBS and incubated with 100 μL/well of the horseradish peroxidase (HRP)-conjugated secondary reagent: anti-mouse IgG (γ chain specific) at 1:1000; anti-sheep IgG (H + L chain specific) at 1:1000; or anti-human IgG (γ chain specific) at 1:2000. All secondary antibodies were purchased from Sigma-Aldrich. The reaction proceeded for 1 h at RT in the dark. The secondary antibodies were diluted with 1X PBS ELISA dilution buffer (0.1% BSA, 0.05% Tween 20 in 1X PBS, pH 7.4). The plates were washed and then incubated for 30 min at RT (in the dark) with 100 μL/well of 0.015% 3, 3′, 5, 5′-tetramethylbenzidine dihydrochloride, 0.01% H_2_O_2_ in 0.1 M citrate buffer, pH 4 solution. Then, 100 μL/well of 2 M H_2_SO_4_ was added to terminate the color development. The absorbance was measured at 450 nm using a UV_max_ multiplate spectrophotometer (Molecular Devices, San Jose, CA, USA).

### 4.4. Reactivity of Anti-Z-DNA Antibodies and SLE Plasmas with DNA Coated at Various Concentrations

MC, EC, CT, HP, SS and lambda DNA were plated onto Immunlon 2HB plates and incubated overnight. Poly(dG-dC) and Br-poly(dG-dC) were also plated to serve as negative and positive controls, respectively, to verify the activity and specificity of the anti-Z-DNA antibodies. Initial stock solutions of DNA were prepared at 5 μg/mL with subsequent 2-fold dilutions. All DNA samples were diluted with 1X SSC. No DNA control wells (1X SSC alone) were also included. After overnight incubation, the samples were incubated with anti-Z-DNA antibody preparations (monoclonal antibody at 1:1500, polyclonal antibody at 1:1500 and SLE plasma at a dilution of 1:1000). The rest of the assay was performed as described above.

### 4.5. Titration of Anti-Z-DNA Antibody Preparations with Fixed Concentrations of DNA

The reactivity of anti-Z-DNA antibodies to fixed amounts of plated DNA (MC, EC, CT, HP, SS and lambda, Br-poly(dG-dC) and poly(dG-dC) at 2.5 μg/mL) was measured by ELISA. The ELISA was performed as described above. DNA diluted in 1X SSC was plated overnight at 4 °C and then incubated the next day with 2-fold serial dilutions of antibody preparations, starting at 1:100 of the monoclonal anti-Z-DNA antibody or polyclonal anti-Z-DNA antibody. On completion of the assay, the absorbance per well was then measured at 450 nm using a multiplate spectrophotometer.

### 4.6. Inhibition of Sheep Anti-Z-DNA Antibody Binding to MC DNA by Br-poly(dG-dC)

The ELISA experiment proceeded as described above with modification. MC DNA was plated overnight (4 °C) at a concentration of 2 μg/mL in 1X SSC. The following day, the plates were blocked for 2 h at RT. During this time, a 1:1 mixture of sheep polyclonal anti-Z-DNA antibody and the inhibitory DNA antigens [Br-poly(dG-dC) or poly(dG-dC)] were prepared. The concentration of polyclonal anti-Z-DNA used in these experiments was dependent on the reactivity of DNA on the plate, as determined in preliminary experiments. For MC DNA detection, the final antibody concentrations were determined to be 1:1000, 1:2000 and 1:4000 (data not shown). Each detection antibody dilution was mixed with solutions of varying concentrations of the inhibitory DNA (Br-poly(dG-dC) or poly(dG-dC). The final concentrations of Br-poly(dG-dC) were 0, 27, 82, 250, 740, 2200, 6600 and 20,000 ng/mL. Poly(dG-dC) served as a control antigen, which should not be inhibitory. The mixes were incubated for 1 h at RT and were prepared to provide sufficient volume for duplicate assay wells. The 1:1 mix solutions were distributed at 100 μL/well, as appropriate, and incubated for 1 h at RT. On completion, the ELISA proceeded as described above.

### 4.7. Binding of Anti-Z-DNA Preparations to Mycobacterium Tuberculosis (MTb) DNA

The ability of monoclonal anti-Z-DNA and polyclonal anti-Z-DNA to bind to MTb DNA was determined by ELISA. DNA (MC, CT, MTb, Br-poly(dG-dC); 2 μg/mL) was plated onto Immulon 2HB plates overnight as described above. The following day, the coated plates were incubated with 100 μL/well of serially diluted monoclonal anti-Z-DNA (starting dilution was 1:100, 3-fold serial dilutions) or polyclonal anti-Z-DNA (starting dilution was 1:100, 2-fold serial dilutions). The protocol was as described above.

### 4.8. Time Course of Antibody Binding to DNA Antigens

This experiment measures by ELISA changes in levels of anti-Z-DNA binding to various sources of DNA over time. Br-poly(dG-dC), CT DNA, EC DNA and MC DNA were plated overnight (4 °C) at 2 μg/mL. The plates were blocked the next day. During this time, all dilutions of antibody (monoclonal anti-Z-DNA and polyclonal anti-Z-DNA) were prepared. Four 3-fold serial dilutions of antibody were used for each DNA antigen. These dilutions were chosen to cover the linear portion of activity of antibody to antigen binding. All reactions were performed in duplicate at 100 μL/well. Assay incubation times were 5, 20 and 60 min. The experiment was designed so that the final detection of antibody occurred at the same time by staggering the addition of anti-Z-DNA antibodies. The ELISA protocol proceeded as described above.

### 4.9. Z-DNA Bioinformatic Analysis

To determine the potential content of Z-DNA regions (ZDR) in the various DNA in this study, we used the *Z-Hunt-II* program, an updated version of *Z-Hunt,* which was downloaded from GitHub [14,48]. This algorithm assesses the thermodynamic propensity of genomic sequences to form left-handed Z-DNA under the influence of negative supercoiling [13]. For each base pair, the program calculates a Z-Score (P_Z_), the ability of a sequence to form Z-DNA, where a higher P_Z_ represents higher propensity to form Z-DNA [14,15].

Z-Potentiality (Z_P_) can be calculated using Equation (1) where β is the size in base pairs (bp) of the total sequence analyzed and P_Z_ is the number of Z-Scores equal to or greater than *x* (see Table 1).
(1)$$Z_{P}=\left(\frac{P_{Z\geq x}\;}{\beta }\right)*100\%$$

As an example, the output of *Z-Hunt-II* for MC DNA is P_Z≥250_ = 142,950, and β = 2,848,904. When entered into Equation (1), the Z-potentiality is calculated as 5.02%.

For this analysis, nucleotide sequences for each DNA were obtained from the National Center for Biotechnology Information (NCBI), the Nucleotide database, and were assessed for the content of Z_P_ at various levels of *x*, including 1, 250, 500, 1000, 2000 and 2500. We used these levels on the basis of reports of others. Ho et al. considered a value of greater than 1 to be consistent with a potential to form Z-DNA, while Herbert, for example, used a threshold of P_Z≥250_ to computationally predict Z-DNA-forming sequences of Alu family members [5]. Table 1 presents values for Z_P_ using various values for *x*.

## 5. Conclusions

These studies provide evidence that monoclonal and polyclonal anti-Z-DNA antibody preparations can be used in immunoassays to detect the presence of Z-DNA in naturally occurring DNA. Along with bioinformatics approaches, immunoassays represent a useful approach to assess the content of Z-DNA in bacterial and viral DNA and its relationship to the induction of immune responses.

## Figures and Tables

**Figure 1 ijms-22-08931-f001:**
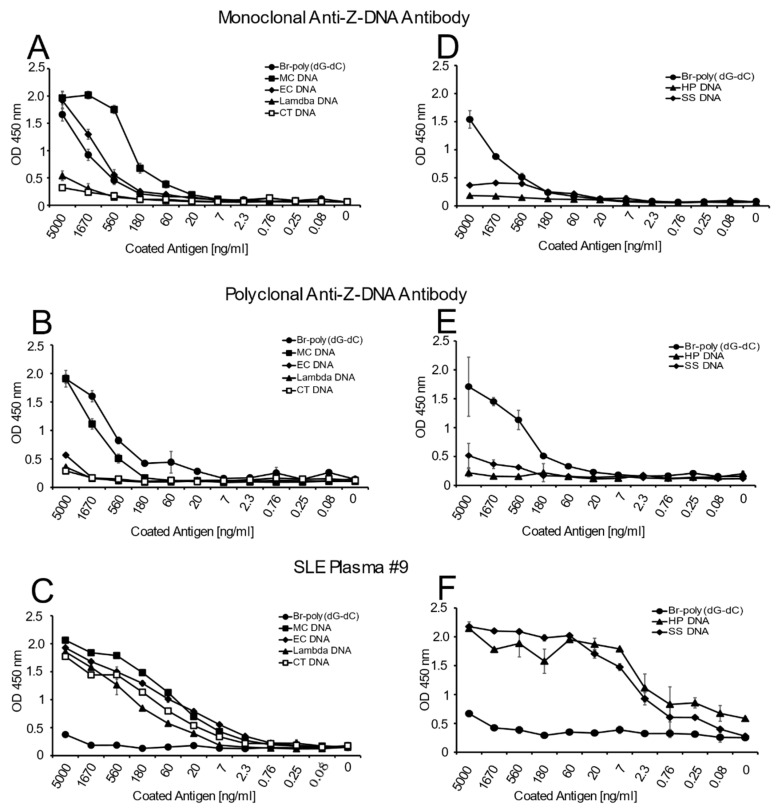
The binding of monoclonal and polyclonal anti-Z-DNA antibodies to natural DNA coated at various concentrations. The binding of either a murine monoclonal anti-Z-DNA antibody (**A**,**D**); 1:1500 dilution), a sheep polyclonal anti-Z-DNA antibody ((**B**,**E**); 1:1500 dilution) or SLE plasma #9 ((**C**,**F**); 1:1000 dilution)) to various concentrations of DNA coated to microtiter plates was determined by ELISA. Brominated poly (dG-dC) (Br-poly(dG-dC)) was used as a positive control and unbrominated poly (dG-dC) was used as a negative control. Results are reported as OD 450 nm values. All treatments were performed in duplicate. Means and standard deviations are shown.

**Figure 2 ijms-22-08931-f002:**
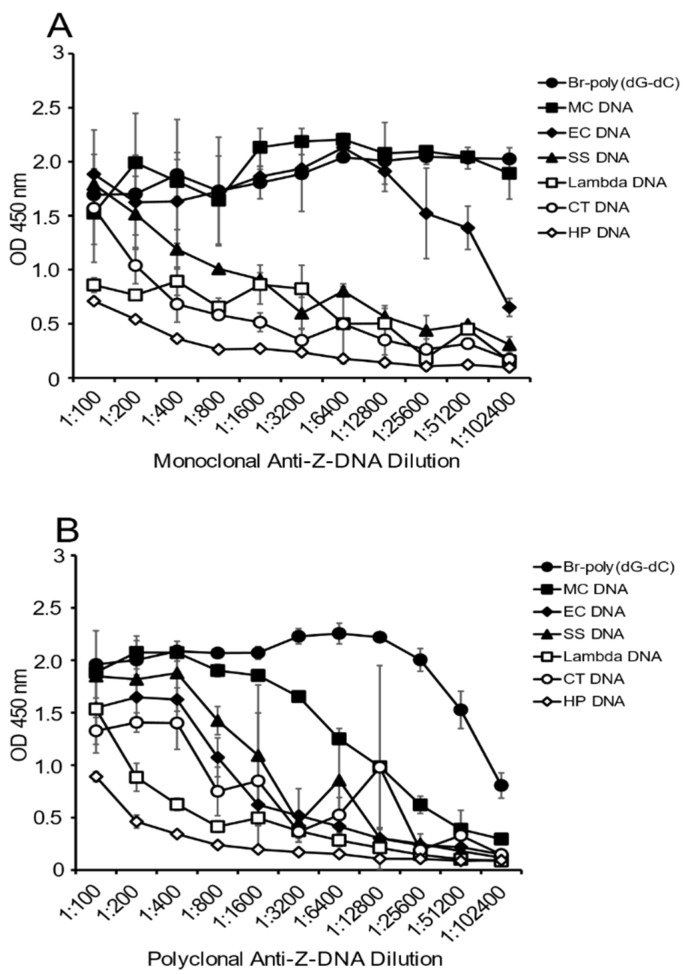
The effect of antibody dilution on the binding of monoclonal and polyclonal anti-Z-DNA antibodies to natural DNA. The binding of various dilutions of monoclonal and polyclonal anti-Z-DNA antibodies to DNA coated at a concentration of 2.5 μg/mL was assessed by ELISA. (**A**) Results for the monoclonal antibody; (**B**) results for the polyclonal antibody. Results are reported in terms of OD 450 nm values. All treatments were performed in duplicate. The means and standard deviations are presented.

**Figure 3 ijms-22-08931-f003:**
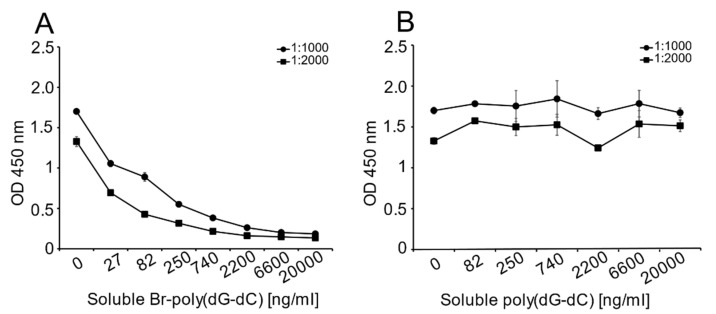
The effects of soluble Br-poly(dG-dC) on the binding of polyclonal anti-Z-DNA antibodies to MC DNA. To show the specificity of the polyclonal anti-Z-DNA antibodies for Z-DNA in MC DNA, the ability of soluble Br-poly(dG-dC) to inhibit antibody binding was tested (**A**). Br-poly(dG-dC) was used a source of Z-DNA; unbrominated poly(dG-dC) was the control (**B**). The binding of the antibody preparations at dilutions of both 1:1000 and 1:2000 was tested. The results are presented as OD 450 nm values. All treatments were performed in duplicate. Means and standard deviations are shown.

**Figure 4 ijms-22-08931-f004:**
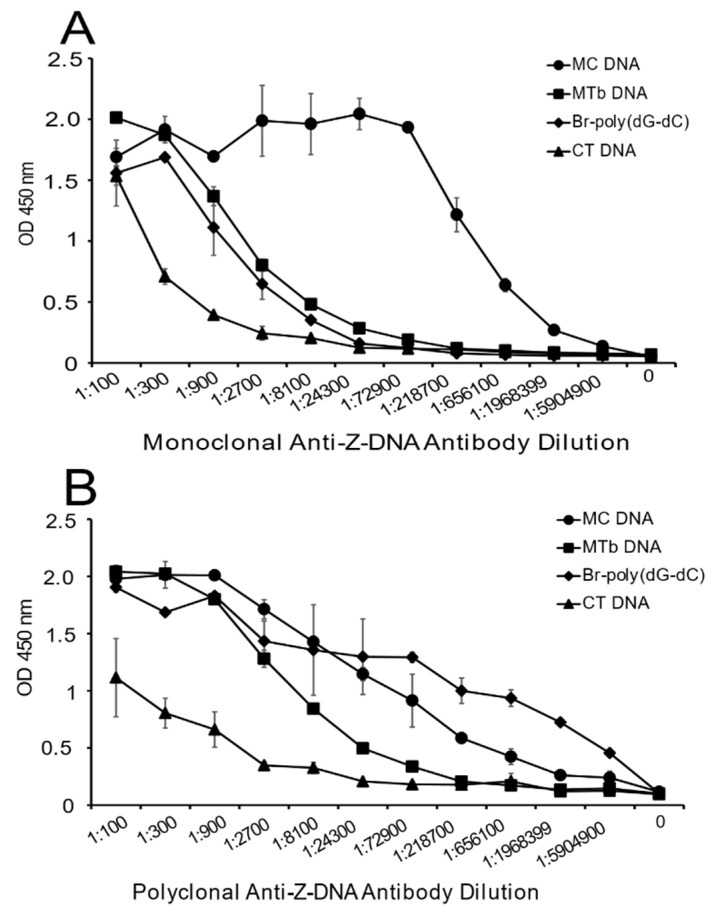
The binding of monoclonal (**A**) and polyclonal (**B**) anti-Z-DNA antibodies to DNA from Mycobacterium tuberculosis. The binding of various dilutions of the anti-Z-DNA antibodies to MTb DNA was assessed by ELISA. All DNA antigens were coated at a concentration of 2 μg/mL. Results are reported as OD 450 nm values. All treatments were performed in duplicate. Means and standard deviations are shown.

**Figure 5 ijms-22-08931-f005:**
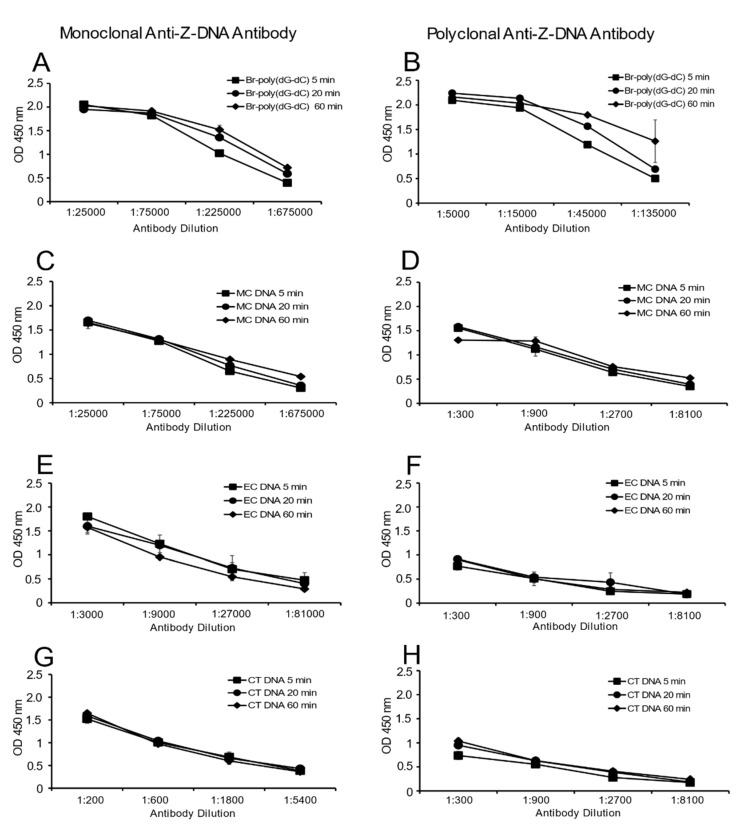
Time course of antibody binding to natural DNA and Br-poly(dG-dC). The binding of monoclonal and polyclonal anti-Z-DNA antibodies to natural DNA (MC and EC) and Br-poly(dG-dC) was tested at 5, 20 and 60 min. (**A**,**C**,**E**,**G**) Results of monoclonal anti-Z-DNA binding; (**B**,**D**,**F**,**H**) polyclonal anti-Z-DNA binding. DNA was coated to plates at 2 μg/mL. Antibody preparations were titered, with the starting concentrations determined by a prior experiment to determine the dilutions covering the linear portion of antibody binding to DNA. Results are presented as the average OD 450 nm values of duplicate wells. Means and standard deviations are shown.

**Table 1 ijms-22-08931-t001:** DNA sources and Z-potentiality.

DNA Source ^a^	NCBI Accession Number	GC% ^b^	Z_P≥1_ (%)	Z_P≥250_ (%)	Z_P≥500_ (%)	Z_P≥1000_ (%)	Z_P≥2000_ (%)	Z_P≥2500_ (%)
*M. luteus*	CP040019.1	73.0%	91.73	5.02	2.91	1.69	0.93	0.76
*M. tuberculosis*	CP008960.1	65.6%	81.62	2.22	1.19	0.68	0.35	0.28
*E. coli*	AE014075.1	50.6%	58.58	0.88	0.44	0.25	0.12	0.09
*λ phage*	DM131435.1	49.8%	57.82	0.58	0.28	0.17	0.09	0.07
Salmon sperm	NC050133.1	43.4%	44.03	0.79	0.60	0.48	0.38	0.09
*H. sapiens*	GRCh38.p13	40.4%	38.11	0.29	0.19	0.12	0.08	0.03
Calf thymus	CM008197.2	41.9%	35.41	0.22	0.15	0.10	0.06	0.02

^a^ The content of Z_P_ at various levels of *x* (1, 250, 500, 1000, 2000 and 2500) was determined for each respective DNA source. The table above provides the seven DNA sources used and the source of the data that were analyzed (NCBI Accession Number). ^b^ GC% content was sourced from the NCBI Genome database. TXIDs are as follows: MC (1270), MTb (83332), EC (562), Lambda (10710), SS (8018), HP (9606) and CT (9913).

## Data Availability

All data will be made available.

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
