# Peer review of "The Binding of Monoclonal and Polyclonal Anti-Z-DNA Antibodies to DNA of Various Species Origin"

_ijms, 2021, doi:10.3390/ijms22168931_

Round 1
Reviewer 1 Report
This study provides a detailed investigation of the abilities of commercially available anti-Z-DNA antibodies to bind non-synthetic naturally existing DNA from different species. This manuscript is well-written. It presents thought-through and meticulously done sets of ELISA-based experiments conducted under the physiological salt condition that detect Z-DNA structure in CG-rich bacterial sequences.
The manuscript has sufficient technical strength and novelty and can be accepted for publication after addressing some concerns.
The primary concern is that the authors generated the brominated control Z-DNA at 150 mM NaCl by contrast with previously published approaches, and this preparation was unstable. Ideally, the independent confirmation of the effectiveness of such a procedure is needed to determine the extent of bromine modification and to evaluate the changes in the ratio of Z- and B- DNAs in such preparations over time. I am wondering if any published studies are addressing the stability of brominated DNA?
It is a detail, but authors might consider adding citations showing how plate coating agents affect the DNA conformation.
Reviewer 2 Report
The authors presents a new method of Z-DNA of various species origin,the paper is well-organized and interesting. However, there are still concerns to be addressed.
1) The method employ the polyclonal anti-Z-DNA antibody to detect the Z-DNA, as we know, the polyclonal antibody has the disadvantage of homogeneity. How to avoid it in the method?
2) please put "Materials and Methods" after the "Introduction".
